# Design and Structural Factors’ Influence on the Fatigue Life of Steel Products with Additive Manufacturing

**DOI:** 10.3390/ma16237315

**Published:** 2023-11-24

**Authors:** Nataliya Kazantseva, Maxim Il’inikh, Victor Kuznetsov, Yulia Koemets, Konstantin Bakhrunov, Maxim Karabanalov

**Affiliations:** 1Institute of Metals Physics, Ural Branch of Russian Academy of Sciences, 620108 Ekaterinburg, Russia; maximilyinykh@gmail.com (M.I.); y.koemets@imp.uran.ru (Y.K.); 2Department of Natural Sciences, Ural State University of Railway Transport, 620034 Ekaterinburg, Russia; 3Department of Heat Treatment and Metal Physics, Ural Federal University, 620002 Ekaterinburg, Russia; wpkuzn@mail.ru (V.K.); m.s.karabanalov@urfu.ru (M.K.); 4National Ilizarov Medical Research Centre for Traumatology and Orthopaedics, 640014 Kurgan, Russia; 5Department of Natural Sciences, Buryat State Agricultural Academy, 670024 Ulan-Ude, Russia; bkk1975@mail.ru

**Keywords:** 3D printing, design, cyclic load, austenitic steel, simulation

## Abstract

The influence of implant design and structural factors on fatigue life under cyclic loading was investigated. The implants were manufactured from 316L steel powder using 3D printing for medical use. A simulation model of implant deformation was built using ANSYS software. The obtained data showed that the geometry of the implant had the necessary margin of safety for osseointegration time. It was found that the stress concentration factor, which is associated with fatigue life, for an implant with a hexagon head and internal thread depends on the mechanical properties of the metal, design, and load conditions. The presence of internal threads and holes in the implant increases the stress concentration factor by more than 10 times. The number of load cycles for the failure of the implant, which was calculated by taking into account a coefficient for reducing the endurance limit, was found to be sufficient for implant osseointegration.

## 1. Introduction

Metallic materials play an important role in medicine. The main studies of the metallic biomaterials associated with different areas include surface treatments, protective covers, the development of new alloys, and new manufacturing [1]. Titanium alloys, CoCr alloy, and 316L low-carbon austenitic steel are well-known biomaterials that have been used for medical activities for many years. Titanium alloys are the most commonly used material in the manufacturing of medical implants and instruments [1,2,3]. Pure titanium and its alloys (Ti-6Al-4V, Ti-6Al-7Nb, Ti-12Mo-6Zr, and Ti-13Nb-13Zr) have excellent biocompatibility, osseointegration, and good elastic properties. However, titanium shows relatively low resistance to tribocorrosion, due to its relatively low stiffness [2]. Developments in new titanium alloys are associated with a possible toxic effect resulting from released vanadium and aluminum in aggressive fluids of the human body [3]. Due to the high ability of titanium to absorb gases (oxygen and nitrogen) when the temperature rises, a disadvantage for titanium implants is the high cost associated with the complexity of their production. Additionally, it was found that the biocompatibility of titanium was associated with its oxides [3]. Titanium alloys are quite resistant to corrosion. However, violation of the oxide protective film on the surface of the implant promotes corrosion, which can cause local inflammation around the implant [1].

Cobalt chromium alloys are also used for surgical implant applications. These biomaterials have superior hardness, strength, and resistance to corrosion [3]. The mechanical properties of these alloys depend on the cast and post-treatment conditions. A fragile sigma phase may form along the interdendritic regions and decrease the plasticity of the alloy material. The corrosion resistance of CoCr alloys depends on their chemical composition and microstructure [3].

316L austenitic steel has good mechanical and corrosive characteristics. The resistance of this steel to pitting corrosion may be improved by adding molybdenum. Resistance to corrosion is lower in stainless steel than in titanium implants [2]. In comparison to titanium alloys, 316L austenitic steel has a lower biocompatibility because of the nickel content and its potential for an allergic reaction. However, unlike titanium implants, the austenitic steel implants have a low production cost [4]. Due to these features, in human surgery, stainless steel is used nowadays for temporary implants only [2]. As for veterinary medicine, steel implants are actively used due to their good mechanical properties-to-cost ratio. The mechanical properties of the 316L austenitic steel depend on the phase content and microstructure. It was found that the presence of nano-scale austenite in micro-scale austenitic grains can improve the ductility of the alloy without much loss of strength [3]. In cast 316L steel, δ-ferrite and some inclusions may be observed. Heat treatments of a cast 316L steel sample substantially influence its microstructure and mechanical properties. This means that the properties of this material can be controlled by changing the production method.

The primary fabrication method of complex and highly loaded metal implants is powder metallurgy [5]. This method includes blending, compaction, sintering, and/or hot isostatic pressing (HIP) processes. Additive manufacturing (3D printing) is also related to powder metallurgy methods. However, unlike usual powder metallurgy, manufacturing using a 3D printer does not use special molds to obtain metal products. Three-dimensional printing uses computer models for production, where digital pictures of real objects can be used. This is especially important for individual medicine. The use of a 3D printer for metal miniature medical products is unique because it allows one to produce finished products with shapes that are not possible to obtain using conventional methods. Three-dimensional printers that use electron beam melting (EBM) with metal powder are considered promising devices for the development of metal implant manufacturing [6]. Unlike laser additive technology, the electron beam melting process of the metal powder occurs in a vacuum and at low speed [7]. This allows for obtaining a more balanced structure and reducing residual tensile stresses in the metal product when compared to products manufactured with a laser 3D printer. The products produced by 3D printers using electron beam melting have a higher surface roughness than those produced by laser 3D printers [7,8,9]. Thus, the EBM method requires more machining and, therefore, a margin regarding the size of the original model used for the final additive manufactured (AM) product. However, in medicine, the roughness of a product’s surface is useful for the osseointegration process [10,11]. In addition, osseointegration is positively impacted by the internal porosity of AM products, which is usually viewed as undesirable for AM metal products in other industries due to a decrease in mechanical properties [11]. The fatigue life of AM metal samples is the object of the different works [12,13,14,15]. In [13], it was shown that the fatigue strength of an AM 316L sample approached that of the corresponding wrought material when subjected to principal stresses aligned with the building plane. However, in a review of the main factors affected by the fatigue life of AM metals, it was shown that those metals produced by AM showed a much lower fatigue strength and greater dispersion than conventionally produced metals because of defects, such as porosity, surface roughness, and lack of melting, which were very effective in reducing fatigue strength [14].

It was found that the fatigue fractures in an AM 316L sample depended on the defects and were initiated by multiple surface defects rather than internal porosity. In addition, it was shown that the fatigue life of AM 316L steel samples depended on the build orientation of the sample during 3D printing and the process parameters of 3D printing [15,16,17]. Physiological environment experiments showed that SLM 316L SS outperformed conventional wrought 316L SS in terms of corrosion resistance and biocompatibility [18]. All of this means that it is much better to use AM products when these parts cannot be obtained in any other way and when the disadvantages of this technology can be minimized.

By using additive manufacturing, implants with a complex geometric shape can be manufactured for individual patient characteristics; this has great potential in medical manufacturing. For the development of new implant design solutions using additive manufacturing, it is necessary to control the functional properties and strength of the AM products. Such tests are a challenging task due to the need for special test benches. AM medical products that are not just complex but also small, as well as having internal threads, are particularly challenging for this type of test. In veterinary medicine, where miniature implants are used, the influence of the implant design on its operational properties and long-term durability is particularly significant. 

A simpler option for testing products that have complex shapes may be to simulate their behavior under operating conditions. Sometimes, in the case of additive manufacturing, a simulation process may be the only solution to a very complicated problem. The topological optimization of AM products is a very interesting subject for research [19,20]. However, the simulation of the fatigue process, along with the study of the structure of a real AM product for a complex shape, is not common. This is especially important for predicting the fatigue properties of AM metal products for medicine since, unlike the technical industry, replacing such products in the event of their destruction is a complex and painful operation.

Usually, the Wohller equation is used to describe the process of fatigue failure in metal. This equation describes the relationship between the stress level, *σ*, and the number of load cycles, *N_f_* [21]:(1)$$\;\sigma^{m_{f}}\cdot N_{f}=C_{f}.$$

The exponent coefficient, *m_f_*, characterizes the slope of the fatigue curve, and the constant *C_f_* determines the position of the fatigue curve on the *x*-axis. In real products, many factors influence fatigue life assessment, so to calculate fatigue resistance, a coefficient for reducing the endurance limit (*K*), which includes the influence of all factors on fatigue resistance, is introduced:(2)$$\sigma_{fp}=\frac{\sigma_{f}}{K},$$

There is $\sigma_{fp}$ fatigue resistance in the product, and there is $\sigma_{f}$ fatigue resistance in the smooth polished reference. The values of $\sigma_{fp}$ are usually less than the average value of the endurance limit of smooth laboratory polished samples, which only characterize the properties of the material [22]. For metal implants, as modifying factors, it is necessary to consider the effects of stress concentration, surface roughness, and chemical and structural homogeneity. This provides a need for structural studies of AM steel products. It is especially important for porous materials because the initiation of a crack inside the material may occur from casting pores or shells [23]. Another variation of the internal fracture may occur when the crack starts around inclusions or hardening particles, which are connected to the inhomogeneity of the material. Such internal defects contribute to an increase in the local concentration of stress in the material [24]. In this case, it is important to know the size of the inclusions or pores, which also influence the fatigue life of the AM metal products.

The purpose of this study is to analyze the effects of the design and structural factors on the fatigue life of a 316L steel implant manufactured by electron beam additive technology.

## 2. Materials and Methods

The implants were manufactured by using the electron beam melting (EBM) method and 316L medical steel powder. The chemical composition of the 316L powder is shown in Table 1.

Arcam A2X EBM equipment (GE Additive Company, Boston, MA, USA) was used. The manufacturer’s recommendations were used to select the 3D printer regime for printing a steel product. The design of the AM implant was chosen in accordance with the RF patent [25]. The implant’s threaded rod had a hexagon head and internal thread. Antiseptic agents can be introduced through holes and a through channel inside the implant. The length of the implant was about 4.5 sm. Additive manufacturing is the only way to produce such a geometric shape for this type of implant.

The surface roughness of the AM samples was measured with a WYKO NT1100 (“VEECO”, Munich, Germany) instrument using the vertical scanning interferometry (VSI) method. The VISION software package (https://www.visionsoftwareonline.com/) was used to determine the roughness parameters Ra and Rz. A ZEISS CrossBeam AURIGA (Carl Zeiss NTS, Oberkochen, Germany) scanning electron microscope (SEM) was used to investigate the microstructure. The simulation process was completed using ANSYS finite element software (https://www.ansys.com/academic). First, the simulation of the fatigue tests was conducted using a simple cylinder with a maximum load of 10 kN, a stress ratio of R = −1, and a frequency of f = 10 Hz, and then the simulation of the implant with a complex shape was carried out. A hexahedral mesh was used for constructing a computational mesh on a 3D model of a standard sample. A mesh based on a linear tetrahedron was used to construct a computational mesh on a 3D model of the implant. The quality of the constructed mesh was checked using the tools built into ANSYS Meshing. The boundary condition Fixed Support was used: rigid embedding. Grid convergence was not carried out in this work. The results of the mechanical test of the conventional steel 316L sample, which was used in the simulation of the cyclic mechanical tests, were taken from [26]. The mechanical properties of 316L steel are presented in Table 2 [26].

## 3. Results

### 3.1. Computer Simulation

#### 3.1.1. Computer Simulation of Simple Cylinder

The loading scheme was chosen in accordance with the conditions for the usual fixing of the implant using the Elizarov apparatus (Figure 1a); the rabbit impact force diagram is shown in Figure 2b. No animals were used for this study.

Figure 2 presents the schema of the sample load used for the cyclic simulation. The left end of the sample has a rigid embedment; a tensile force is applied to the right end. The simulation load was carried out at room temperature.

Modeling allowed us to determine the dangerous point (region) in the sample (product), in which the process of destruction under high-cycle fatigue is localized. Figure 3 shows the results of the simulation in ANSYS software (https://www.ansys.com/academic), where *N* is the number of cycles to failure, and *σ*_*a*_ is the strain amplitude.

The theoretical stress concentration factor in the deformed region was determined as follows [27]:(3)$$\alpha_{a}=\frac{\sigma_{imax}}{\sigma_{i\infty }},$$
where $\sigma_{imax}\;$ is the equivalent maximum local stress, and $\sigma_{i\infty }$ is the equivalent stress without stress concentration. As can be seen from Figure 3, the maximum stresses are concentrated in the middle of the sample. With decreasing stress, the deformation zone becomes narrower, and the number of cycles before failure increases. This shows that the process of damage accumulation during high-cycle fatigue leads to the destruction of structures localized in a small area. Figure 3 also shows that when the number of cycles of failure increases up to 1,000,000, destruction occurs under the stress value, which is less than the yield strength.

The fatigue limit for 10,000,000 cycles obtained from the graphs for the reference sample [26] was *σ**_r_* = 146.45 MPa, and for the simulated sample, it was *σ**_r_* = 146.28 MPa (Table 3 and Table 4).

The Basquin fatigue strength equation between fatigue life and applied load presents the function relationship $y=a\cdot x^{m}$. The empirical coefficient *m* is included in the Basquin fatigue strength equation and shows the limited endurance of the studied sample [28]. The coefficient *m* can be calculated from the slope value of the S–N fatigue curve using logarithmic co-ordinates:(4)$$N_{f}\cdot \sigma_{r}^{m}=const.\;$$

In our case, *m* = 0.11. Figure 4 shows the S–N curves, demonstrating good convergence between the chosen model and the reference.

#### 3.1.2. Computer Simulation of the Implant

We used the obtained data from a simple cylinder simulation to further evaluate the fatigue life of the implant. In order to simulate the cyclic loading of the implant, we rigidly fixed the left edge of the hexagon screw (Figure 5).

Then, on a flat surface (shown in red in Figure 6a), we normally applied a force of 30 N. The maximum load was determined by the force magnitude applied by the rabbit (30–40 N). The calculation of the equivalent stresses, according to von Mises, showed that the maximum stress value of 305.3 MPa was observed at the transition of the thread into the thrust flange (Figure 6b). This value is lower than the tensile yield strength of the 316L steel (Table 2). The theoretical stress concentration factor after the failure of the implant was calculated as *α_a_* = 1.73. Thus, this factor shows the influence of the design of the implant on the value of the resulting stress.

Further, we increased the load to 40 N, changed the loading pattern, and applied force to the entire screw (Figure 7). The maximum stress value, $\sigma_{adm}=273.3$ MPa, was calculated at the transition of the thread into the thrust flange (Figure 7). The theoretical stress concentration factor after the failure of the implant was calculated as *α_a_* = 1.55. In this case, the change in the loading condition leads to a decrease in the theoretical stress concentration factors and an increase in fatigue life.

As was shown by video surveillance, a Chinchilla rabbit hit its paw 12 times for 30 s in an 8 h period while awake. Implant osseointegration time was found to be 6 months. The number of rabbit paw strokes over 6 months was *N* = 37,800.

The fatigue safety factor may be calculated as follows [24]:(5)$$F_{s}=\frac{\sigma_{adm}}{\sigma_{a}},$$
where $\sigma_{adm}$ is the admissible stress, and $\sigma_{a}$ is the alternating stress applied. According to [29], the fatigue safety factor, $F_{s}\geq 1.5$, is required to ensure a conservative level of reliability, which should lead to infinite life. Safety factors of $1\leq F_{s}<1.5$ imply marginal design, for which the confidence level is not adequate for the infinite life requirement. A safety factor below 1 is unacceptable. In our case, the safety coefficient was *F_s_
*= 1.54.

Thus, the analysis of the obtained data showed that the geometry of the implant had the necessary margin of safety for the osseointegration time.

### 3.2. Structural Study

The coefficient for reducing the endurance limit (*K*) with integrated consideration of the factors may be calculated as follows [22]:(6)$$K=\frac{K_{\sigma }}{K_{d}}+\frac{1}{K_{f}}-1,$$
where $K_{\sigma }$ is the effective stress concentration factor, $K_{d}$ is the scale factor, and $K_{f}$ is the coefficient of the surface roughness influence.

The effective stress concentration factor *K*_σ_ may be calculated by the following formula [22]:(7)$$K_{\sigma }=1+q\cdot \left(a_{a}-1\right),$$

The theoretical stress concentration factor depends only on the shape of the part and does not depend on the material from which the part is made. On the contrary, the material sensitivity coefficient for stress concentration, 0<q<1, depends mainly on the properties of the metal. For low-carbon steel, q≈0.2–0.4 [22].

Thus, the effective stress concentration factor for a studied implant, calculated by using Formula (7), is *K_σ_* = 1.22.

The scale factor ($K_{d}$) depends on the size of the part and, in the case of a part with a diameter of less than 10 mm, can be taken as 1. The coefficient of the surface roughness influence ($K_{f}<1$) depends on the surface finish quality.

In order to check if a factor that takes into account the heterogeneity of the chemical composition of the studied AM steel implant is needed, we conducted a study using the ZEISS Cross Beam AURIGA scanning electron microscope (SEM) (Carl Zeiss NTS, Oberkochen, Germany) equipped with an EBSD HKL Inca spectrometer that used an Oxford Instruments Channel 5 analyzing system (Hobro, Denmark). The results of the analysis of the chemical composition of the implant are presented in Figure 8 and in Table 3. 

We conducted a chemical analysis of various areas of the implant, in Figure 8 these areas are indicated by the letters A, B, C. The sizes of the areas under study are pointed in different colors, area A is pointed in pink, B is pointed in yellow, C is in blue. Below in Figure 8 we present the enlarged images of areas A, B and C, in which the areas where chemical analysis was performed are outlined in pink.

According to the obtained results, it can be seen that the studied AM steel implant has a uniform chemical composition. Thus, we cannot take into account the chemical composition factor as a modifying factor in the calculation of the coefficient for reducing the endurance limit (*K*).

The density of the AM steel implant (7.75 g/mm^3^, 98.2%) was measured by using the Archimedes method. The lower density of the AM material indicates the presence of internal pores. Figure 9 presents the cross-section of the AM steel implant. Two different types of defects were found in the structure of the implant, such as small gas pores of about 1 μm in size and technological pores, which formed due to the lack of fusion (Figure 9; referred to with arrows). The technological pores have sharp edges that are irregular in shape and have powder inside. The areas of the internal thread of the implant also have deep, sharp edges with an irregular shape (Figure 9).

Figure 10 presents the surface structure of the different regions of the implant. As can be seen from the figure, the surfaces of the implant parts have different levels of roughness, which is associated with the orientation (tilt angle) of the different regions of the implant to the build direction inside the 3D printer chamber during manufacturing. The surface profiles of the AM steel implant for different parts are presented in Figure 11.

A comparison of the surface roughness parameters for the different parts of the AM steel implant is shown in Table 4.

The arithmetic roughness average of the surface (*R_a_*), the average height of the largest irregularities (*R_z_* ≈ 4*R_a_*), the root-mean-square of the surface roughness (*Rq*), and the height of the single deepest valley in a surface trace (*Rt*) were calculated as follows:(8)$$R_{a}=\frac{1}{n}\cdot \overset{n}{\underset{i=1}{\sum }}y_{i},$$
(9)$$R_{z}=\frac{1}{4}\cdot (\overset{4}{\underset{i=1}{\sum }}\left|y_{imin}\right|+\overset{4}{\underset{i=1}{\sum }}\left|y_{imax}\right|),$$
(10)$$R_{q}=\sqrt{\frac{{\left(y_{i}-R_{a}\right)}^{2}}{n}.}$$

All values measured in the critical region (B) were found to be smaller than those in region A. This fact is associated with the tilt of the part of the implant (B) to the direction of the build during manufacturing (Figure 8).

The coefficient of the surface roughness influence ($K_{f}$) may be calculated as follows [22]:(11)$$K_{f}\approx 1-0.22\cdot \log R_{z}\cdot \left(\log\left(\frac{\sigma_{b}}{20}\right)-1\right),$$
where $R_{z}$ is the average height of the largest irregularities on the surface of the part, and $\sigma_{b}$ is the ultimate strength. In our case, for the critical region (B), $K_{f}\;$= 0.94.

Thus, the coefficient for reducing the endurance limit of the critical region (B) of the studied steel implant was calculated by using Formula (6): *K* = 1.47. In this case, when taking into account the fact that the stress level, *σ*, and the number of load cycles, *N_f_*, are nearly linearly dependent, the maximum stress value, $\sigma_{adm}=273.3$ MPa, will correspond to the number of the cycles for the failure of the implant, *N*
≈ 42858; this is also enough for implant osseointegration.

## 4. Discussion

It is known that the design of metal parts is an essential parameter that directly alters fatigue life. Using the topology optimization of the AM products allows for a decrease in weight while maintaining all mechanical properties. This idea is now widely used in the automotive and aerospace industries, as it allows one to obtain unusually shaped products that cannot be produced in a conventional way [19,20]. In the case of screw connections, the complex design is characterized by a stress concentration factor. The value of the stress concentration factor depends on the mechanical properties, part geometry, and surface properties of the metal. In order to reduce stress concentrations, it is necessary to ensure the smooth transition of internal stress flow between the product areas [27]. As is shown in our research, the theoretical stress concentration factor, which is included in the formula for the effective stress concentration factor, also depends on load conditions. The complicated shape of the implant with internal threads and holes increases the theoretical stress concentration factor by more than 10 times in comparison to a simple reference cylinder sample.

The fracture of the product under cyclic loads always occurs suddenly and begins with the formation of microcracks, which move deeper into the material and reduce the cross-sectional area of the part. It was found that surface roughness and internal defects also have effects on the fatigue life of an AM metal product [15,19]. Because of the internal defects (pores), the statistical data from fatigue tests of AM 316L steel samples show greater dispersion and lower fatigue life than that of a conventional steel sample reference [30]. Among the internal defects (pores) in AM metals, the most dangerous are technological pores, which have sharp edges. Sharp edges act as stress concentrator points and may provoke the formation of cracks. In our case, technological pores with sharp edges were observed inside the implant (Figure 9). This means that an area with such technological pores may have a shorter lifetime than the areas without pores.

Post-processing, machining and HIP (hot isostatic pressing) are usually recommended for AM metal samples to decrease the porosity and surface roughness and increase fatigue life. However, as was shown in [16], unlike Ti6Al4V and CoCr, the AM 316L steel samples were found to be highly defective and residual stress-tolerant. The HIP and stress-relieving heat treatments of AM 316L steel samples were unable to improve their fatigue strength. Thus, the only option to reduce porosity may be to adjust the printing parameters. As was found in [31], electron beam melting allows for samples with a density of up to 99.8%. As shown by Zhang et al., there is a critical size (more than 50 µm) regarding the technological pores involved in the destruction process under high cyclic loads [32]. At low-load levels (less than 275 MPa), fatigue destruction in AM 316L steel begins predominantly from the surface [33].

Surface roughness is believed to promote the osseointegration of implants [10]. In our case, we found a relatively low level of surface roughness regarding the studied steel AM implant (Table 4). The calculated coefficient of the influence of surface roughness was within acceptable limits. However, the implant surface had sharp edges, which may negatively affect fatigue life (Figure 9). This means that additional surface post-processing is required for AM steel implants. Different surface treatments are commonly performed on medical implant materials to promote corrosion resistance and biocompatibility. However, any machining, laser melting, or ultrasonic treatment significantly increases the cost of the production process and may be generally unacceptable in the case of small AM implants with complex shapes. In this case, the best option may be to use electro-polishing, which smooths the surface and performs surface passivation. Shih et al. [34] found that surface passivation by anodic polarization, which produced an amorphous chromium and iron oxide layer in a 316L steel sample, increased the corrosion resistance of this sample in Ringer’s physiological solution at 37 °C. It was also found that the amorphous oxide layer possessed excellent antithrombotic characteristics for surface protection in 316L stainless samples [35].

## 5. Conclusions

This work presents a method for the assessment and prediction of the fatigue life of metal implants that have complex forms before their use in surgery. The obtained results can be summarized as the following:The simulation of the deformation process showed that the geometry of the implant had the necessary margin of safety for the osseointegration time.It was found that the stress concentration factor, which is associated with fatigue life, for an implant with a hexagon head and internal thread depends on the mechanical properties of the metal, design, and load conditions. The presence of internal threads and holes in the implant increases the theoretical stress concentration factor by more than 10 times.The calculated coefficient of the influence of surface roughness is within acceptable limits. Implant surface and technological pores with sharp edges, which may negatively affect fatigue life, were observed in the AM 316L steel implant.In order to increase the fatigue life of the implant, a more careful selection of manufacturing modes and the use of electro-polishing as a post-processing process may be recommended.The number of cycles for the failure of the implant, *N*
≈ 42858, which was calculated taking into account a coefficient for reducing the endurance limit, is enough for implant osseointegration.

## Figures and Tables

**Figure 1 materials-16-07315-f001:**
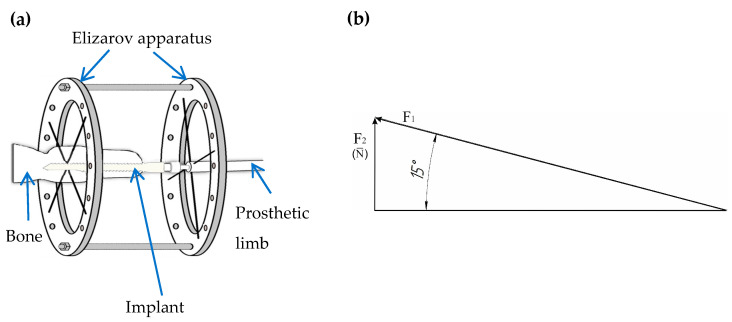
The loading scheme: (**a**) the schema of implant setting in the bone of a Chinchilla rabbit in the external implant fixation device; (**b**) the rabbit impact force diagram, where F_1_ = 150 N—rabbit impact force; F_2_ = 40.2 N—force projection on the implant axis ($\bar{N}$).

**Figure 2 materials-16-07315-f002:**
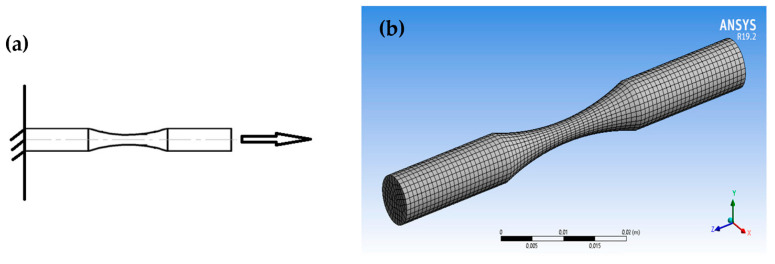
Schema of the load (**a**) and 3D finite element mesh (**b**) of the steel samples.

**Figure 3 materials-16-07315-f003:**
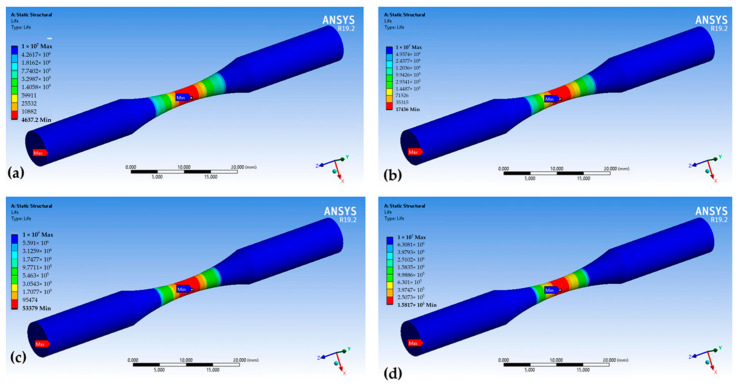
Results of the cyclic mechanical test simulation of the studied steel sample in ANSYS: (**a**) *N* = 4637, *σ_a_* = 332.98 MPa; (**b**) *N* = 17,436, *σ_a_* = 302.84 MPa; (**c**) *N* = 53,379, *σ_a_* = 277.56 MPa; (**d**) *N* = 158,170, *σ_a_* = 242.27 MPa; (**e**) *N* = 390,330, *σ_a_* = 227.13 MPa; (**f**) *N* = 972,140, *σ_a_* = 176.56 MPa; (**g**) *N* = 4,265,800, *σ_a_* = 161.42 MPa; (**h**) *N* = 7,468,900, *σ_s_* =151.42 MPa; (**i**) *N* = 9,438,400, *σ_a_* = 146.28 MPa.

**Figure 4 materials-16-07315-f004:**
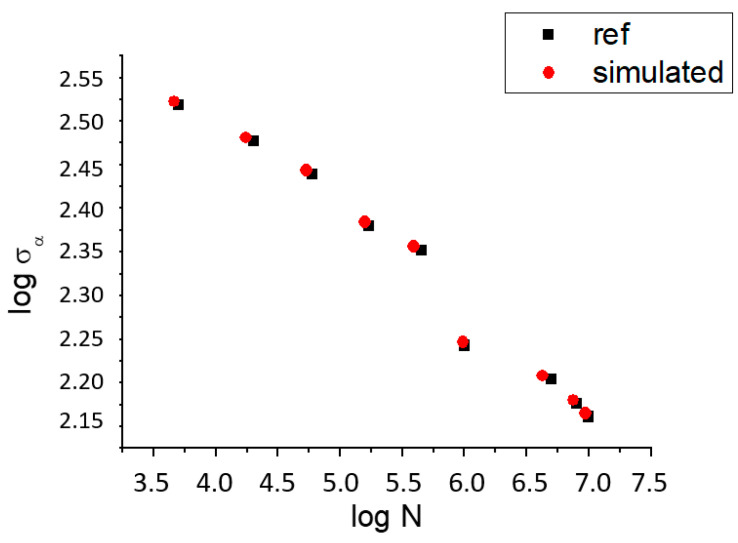
Stress-life (S–N) curves for the reference and simulated samples.

**Figure 5 materials-16-07315-f005:**
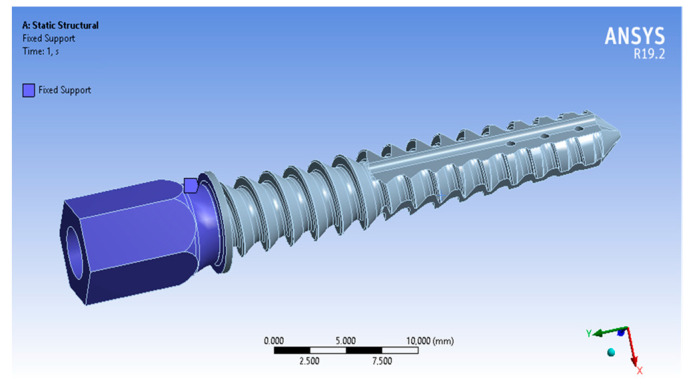
The schema of steel implant fixture for the cyclic mechanical test.

**Figure 6 materials-16-07315-f006:**
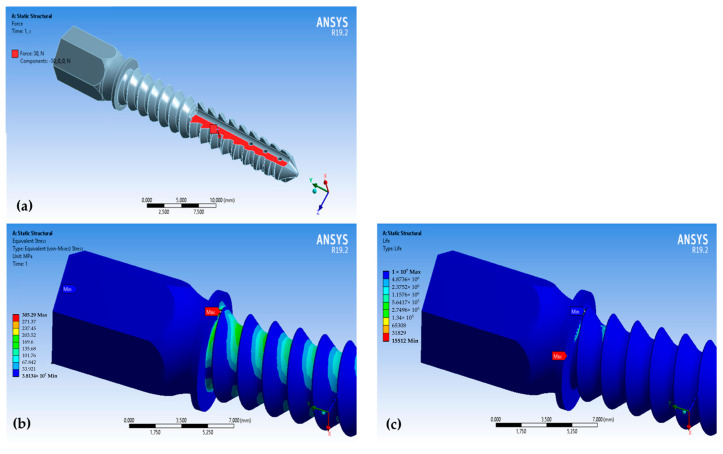
The results of the cyclic simulation of the steel implant: (**a**) the schema of the load of the steel implant; (**b**) the von Mises stress distribution in the steel implant under a load of 30N; (**c**) the distribution of the number of cycles to failure of the implant: *N* = 15,512.

**Figure 7 materials-16-07315-f007:**
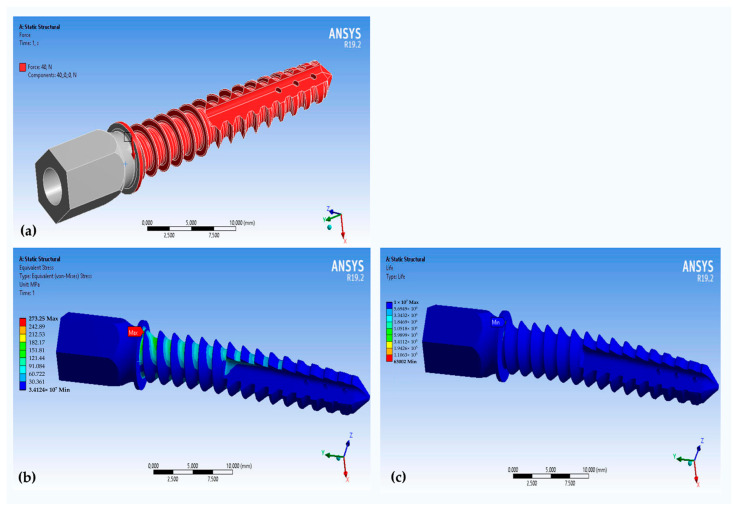
The results of the cyclic simulation of the steel implant: (**a**) the schema of the load of the steel implant; (**b**) the von Mises stress distribution in the steel implant under a load of 40N; (**c**) the distribution of the number of the cycles to failure of the implant, *N* = 63,002.

**Figure 8 materials-16-07315-f008:**
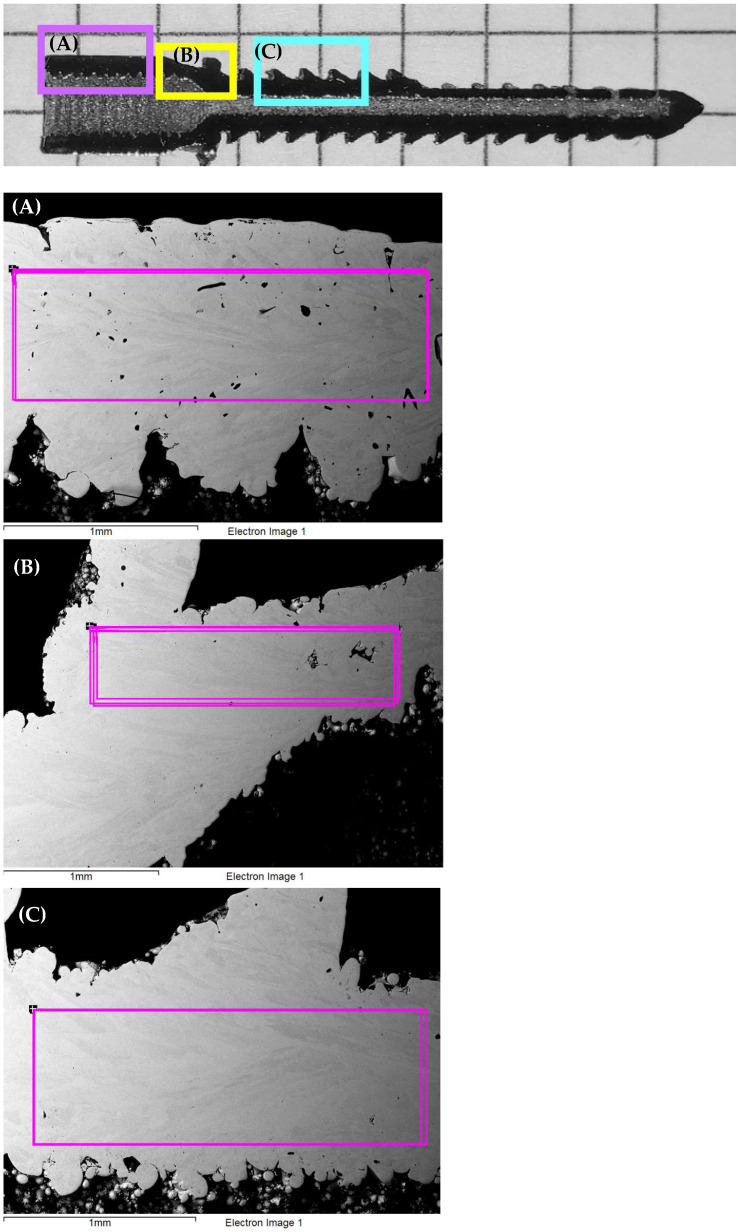
Cross-section of the implant with different pointed regions, where the chemical compositions were measured: SEM results: (**A**)—enlarged images of area A; (**B**)—enlarged images of area B; (**C**)—enlarged images of area C.

**Figure 9 materials-16-07315-f009:**
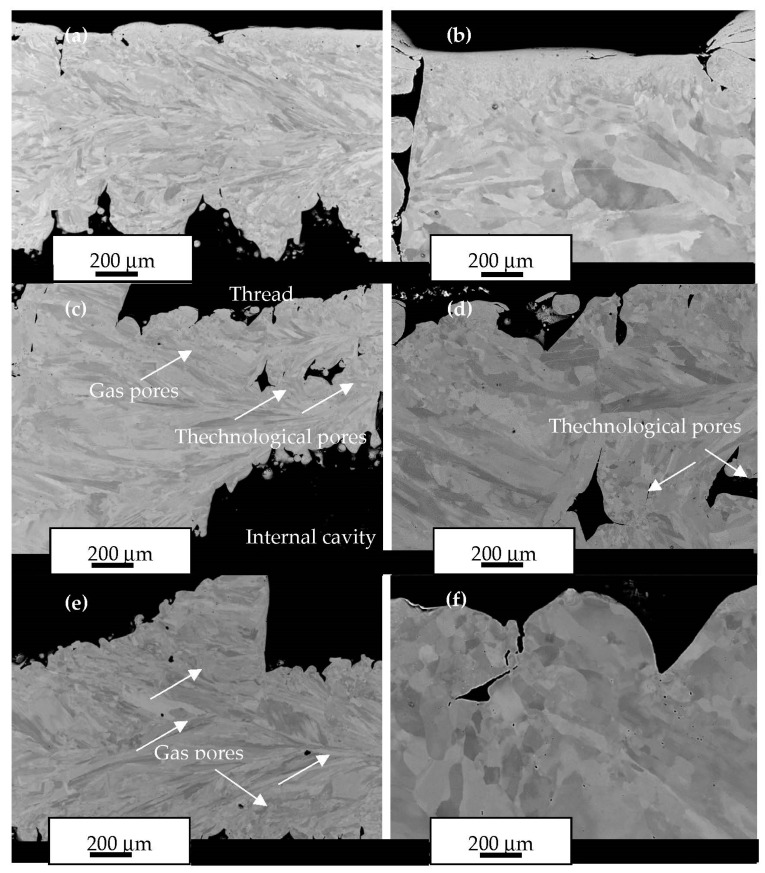
Structure of the AM steel implant in different regions; SEM images of (**a**) region A; (**b**) the same as (**a**) with higher magnification; (**c**) region B; (**d**) the same as (**c**) with higher magnification; (**e**) region C; (**f**) the same as (**e**) with higher magnification.

**Figure 10 materials-16-07315-f010:**
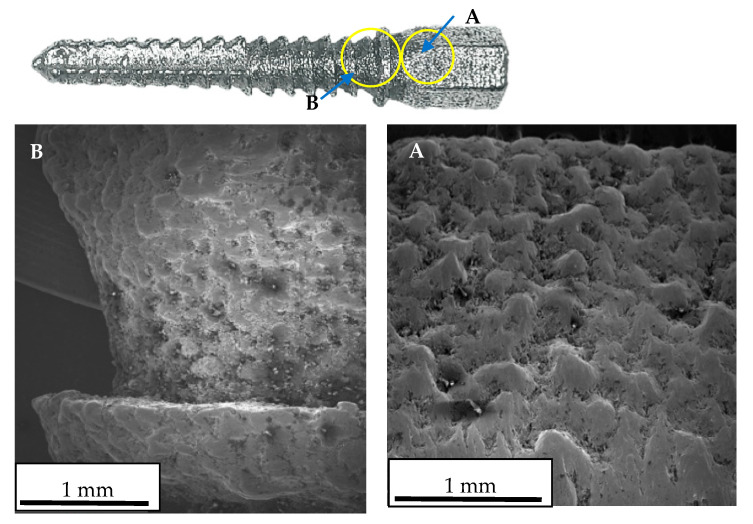
Surface of the AM steel implant for different parts: SEM images: (**A**)—region A with higher magnification: (**B**)—region B with higher magnification.

**Figure 11 materials-16-07315-f011:**
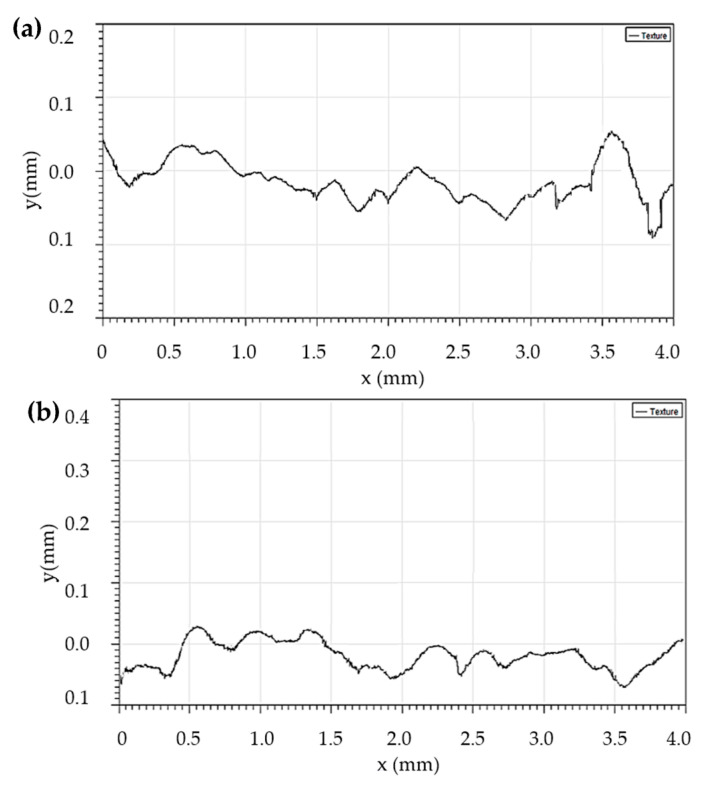
Surface profiles of the AM steel implant for different parts; VSI results: (**a**)—region A; (**b**) region B.

**Table 1 materials-16-07315-t001:** Chemical composition of investigated alloys: wt. %.

Alloy/Element	Fe	Cr	Ni	Mn	Si	Mo
Steel 316L	base	17.28	11.11	1.22	0.78	2.85

**Table 2 materials-16-07315-t002:** Mechanical properties of 316L steel [25].

Yong Modulus, E, GPa	Poisson’s Ratio, μ	Tensile Yield Strength, *σ*_0.2_, MPa	Ultimate Strength, *σ_b_*, MPa
165	0.3	332	673

**Table 3 materials-16-07315-t003:** Results of chemical composition in the different regions of an AM 316L steel implant: wt.%.

Region/Element	Fe	Si	Cr	Mn	Ni	Mo
A	66.7	0.8	17.3	1.2	11.2	2.8
B	66.6	0.9	17.3	1.3	11.1	2.8
C	66.9	0.7	17.3	1.2	11.0	2.9

**Table 4 materials-16-07315-t004:** Results of roughness measurements of AM 316L steel implant: μm.

	Region A	Region B
Ra	33	15
Rz	84	60
Rq	36.5	12.5
Rt	80	57

## Data Availability

Data are available upon request.

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
