# Peer review of "Design and Structural Factors’ Influence on the Fatigue Life of Steel Products with Additive Manufacturing"

_materials, 2023, doi:10.3390/ma16237315_

Round 1
Reviewer 1 Report (Previous Reviewer 2)
Comments and Suggestions for Authors
This manuscript has been greatly improved, with several tiny issues remaining.
1. Clearly label a and b in Figure 1, Figure 2,
2. Although sa and N are standard names, the meaning of these symbols should still be provided clearly in the caption.
Figure 9: Which one is the gas pore, and which one is the technical pore? Use arrows with different colors to point out these pores
Author Response
Many thanks for your comments. We corrected all.

Reviewer 2 Report (New Reviewer)
Comments and Suggestions for Authors
-Implants are made from different bio-med materials specially Titanium. Do a proper survey on different materials used by previous researchers and what problems or complications were found during use.
-Literature gap is not stated.
-Define the objectives clearly.
-Discuss about the testing standards used.
-Where is the boundary conditions? Meshing? Discuss about the convergence test for choosing mesh size.
-Provide big and clear figs.
Author Response
Dear Reviewer, many thanks for your comments. We corrected all.

This manuscript is a resubmission of an earlier submission. The following is a list of the peer review reports and author responses from that submission.
Round 1
Reviewer 1 Report
Comments and Suggestions for Authors
The aim of the work and its organization are not very clear; therefore, a more systematic description of the performed operations would be desirable. It appears that the manuscript is composed of seemingly separate sections: on one hand, the simulation, and on the other hand, the experimental characterization of a real sample. In fact, the authors investigate the influence of the design of a steel implant solely through simulation, but they also construct the implant without subjecting it to mechanical tests. Moreover, they provide only a superficial characterization of the implant by evaluating roughness and morphology (without any chemical characterization that could potentially affect its mechanical properties). They claim that these aspects influence the mechanical properties of the implant, as demonstrated in the literature. This is just one of the critical aspects of the work, which concludes with findings that are almost trivial: it is already known that "It was found that stress concentration factor, associated with the fatigue life, .... depends on the mechanical properties of the metal, design, and load conditions," and that the presence of pores has a negative effect on fatigue life.
In order to be publishable, the work's outline should be completely rethought, and the manuscript should be rewritten, paying careful attention to its form.
Comments on the Quality of English LanguageModerate editing of English language required
Reviewer 2 Report
Comments and Suggestions for Authors
1. The abstract lacks the emphasis of the novelty, significance, and main results of this study.
2. Line 56-67: The whole paragraph only cited one reference, which is not enough.
3. In the introduction, the author only mentioned ‘implants’. What are the functions of the implants? More information and background about the implants (Figure 1) investigated in this study should be provided.
4. The last paragraph in the introduction should provide a brief introduction of what has been done in this study.
5. What is the reason for conducting simulation for simple cylinders?
6. The comparison relationship between the results of simple cylinders and implants is not established clearly. Only in the discussion section (Line 236), do the authors conduct a slight comparison. More comparison and demonstration should provided in the results section.
7. The meaning of the symbols used in Figure 3 should be provided in the caption and demonstrated very clearly in the main text. Actually, the demonstration of Figure 3 is not clear and comprehensive. (Line 143)
8. The meaning of the symbols used in equations (Line 150 and Line 153) should be provided immediately after presenting these equations.
9. Figure 4: Use a decimal point instead of a comma.
10. Why is the roughness in Region A smaller than that in Region B? (Line 211)
11. While the gas pores are pointed out by white arrows, the technological pores are not pointed out in Figure 9.
12. The significance and novelty of this study should be further reinforced in the conclusions.
Comments on the Quality of English LanguageThe writing of this manuscript needs significant improvements. The issues include but are not included the following ones.
1. Line 43: ‘In [9] was shown’ → ‘The study conducted by xx et al.[9] showed that ’
2. Line 46: ‘effected of the fatigue life’ → ‘that can influence the fatigue life’
3. Line 49: ‘melting which were’ → ‘melting, which were’
4. Line 54: ‘much better to use’ → ‘are more suitable be used’
5. Line 66: ‘no one tests of this implant were carried out before the surgical operation.’ → ‘the tests of this implant were not conducted before the surgical operation’
6. Line 74: ‘the Weller equation used for’ → ‘the Weller equation was used for’
7. Line 92: ‘is done in’ → ‘is shown in’
8. Figure 3h: ss or sa ?
9. Line 140: ‘with decreasing stress’ → ‘With the decrement in the stress’
10. Line 199: ‘Without taken’ → ‘Without taking’
11. Line 234: ‘To reduce stress concentrations’ → ‘Reducing stress concentrations’
12. Line 240: ‘are also effects on’ → ‘can also affect’
13. Line 247: lover?
14. Line 275: ‘which associated with’ → ‘which is associated with’
15. Line 276: ‘depends on’ → ‘depending on’
Reviewer 3 Report
Comments and Suggestions for Authors
The first point presented under the conclusion is not supported in the manuscript. The authors simulate a smooth implant in the manuscript and have found the stress values under different loading conditions. However, the real 3D-printed implant has significant levels of surface roughness values and porosities, which contributes massively towards its fatigue life. Without any link between those two, the authors are comparing those two results, which can be highly erroneous.
The first paragraph of the introduction should be rewritten to better represent the idea of the manuscript. Here, the authors discuss several key points without making any significant discussion or details and without proper links among them.
The authors should elaborate on on the "Weller equation" they used on page 74 of page 2. The usual equation in use would be the Wohler equation.
What is the unit of the dimension available in line 102 of page 3?
Authors should provide a detailed mechanical drawing (at least with the dimensions of key features) to understand the severity of the design.
Please elaborate on lines 195 & 196 on page 7.
There are several spelling mistakes. Please read through the manuscript again.
Comments on the Quality of English LanguageThere are several spelling mistakes, please read through the manuscript.
The introduction section should be re-written.